# Deep Hybrid Learning Prediction of Patient-Specific Quality Assurance in Radiotherapy: Implementation in Clinical Routine

**DOI:** 10.3390/diagnostics13050943

**Published:** 2023-03-02

**Authors:** Noémie Moreau, Laurine Bonnor, Cyril Jaudet, Laetitia Lechippey, Nadia Falzone, Alain Batalla, Cindy Bertaut, Aurélien Corroyer-Dulmont

**Affiliations:** 1Medical Physics Department, CLCC François Baclesse, 14000 Caen, France; 2GenesisCare Theranostics, Building 1 & 11, The Mill, 41–43 Bourke Road, Alexandria, NSW 2015, Australia; 3Medical Physics Department, Cherbourg Hospital, 50100 Cherbourg, France; 4Normandie University, UNICAEN, CNRS, ISTCT, CYCERON, 14000 Caen, France

**Keywords:** machine learning, deep hybrid learning, radiotherapy, VMAT, quality assurance, clinical routine

## Abstract

Background: Arc therapy allows for better dose deposition conformation, but the radiotherapy plans (RT plans) are more complex, requiring patient-specific pre-treatment quality assurance (QA). In turn, pre-treatment QA adds to the workload. The objective of this study was to develop a predictive model of Delta4-QA results based on RT-plan complexity indices to reduce QA workload. Methods. Six complexity indices were extracted from 1632 RT VMAT plans. A machine learning (ML) model was developed for classification purpose (two classes: compliance with the QA plan or not). For more complex locations (breast, pelvis and head and neck), innovative deep hybrid learning (DHL) was trained to achieve better performance. Results. For not complex RT plans (with brain and thorax tumor locations), the ML model achieved 100% specificity and 98.9% sensitivity. However, for more complex RT plans, specificity falls to 87%. For these complex RT plans, an innovative QA classification method using DHL was developed and achieved a sensitivity of 100% and a specificity of 97.72%. Conclusions. The ML and DHL models predicted QA results with a high degree of accuracy. Our predictive QA online platform is offering substantial time savings in terms of accelerator occupancy and working time.

## 1. Introduction

The past decade has seen several technological advances that have paved the way for improved radiotherapy techniques. One such technique, volumetric intensity modulated arc therapy (VMAT) [1], works by modulating the beam fluence during a rotation (or an arc) of the accelerator arm to deliver a tumoricidal dose while preserving healthy tissue. As the accelerator arm rotates, the multi-leaf collimators (MLC) must geometrically conform to the dynamic treatment volume while the tumor is being irradiated, making arc therapy treatment plans more complex than conventional treatments [2]. With the increase in plan complexity, there is a huge challenge to accommodate the patient anatomy, dosimetric constraints, optimization algorithms and the linac’s capabilities to ensure that the treatment delivered is as planned on the treatment planning system (TPS). In this context, patient-specific quality assurance (QA) is mandatory and performed for each plan before the patient’s treatment. Using specific phantoms, QA checks whether the planned dose on the TPS corresponds to the dose delivered to the patient on the accelerator using two metrics (mean gamma and gamma index) [3] and thus decides whether or not a treatment plan complies. It is important to note that QA is a time-consuming activity that represents a significant amount of unavailability for the machines to treat patients. Moreover, in our center, only five percent of the treatment plans are non-compliant (gamma index < 0.95, 3%/3 mm).

Several alternatives to QA have been studied and presented in different literature reviews [4,5,6,7,8]. Two main methods were proposed for predicting QA outcomes using complexity indices calculated from treatment plans. One is based on a threshold approach of complexity indices [9]. In this method, QA results are determined according to a decision tree that integrates two complexity indices (combination of Leaf Travel and Modulation Complexity Score (LTMCS) and monitor unit) to eliminate the risk of false positives. As specified [10], false positives (prediction is conforming QA when in reality QA is non-compliant) are equivalent to Type I error, while false negatives are equivalent to Type II error. This last point is still important to take into account in the prediction results because a high number of type II errors requires a lot of QA on the machine. Therefore, the results obtained with the decision tree show a large number of type II errors for complex locations, such as the breast [9].

The second method is a more complex approach based on prediction models such as machine learning regression and classification. As presented in the recent review “Integration of AI and machine learning in radiotherapy QA” [11], different studies have been realized on the use of AI for patient QA in radiotherapy. However, except for the studies by Granville et al. [12] and Hirashima et al. [13], the AI models were created on few numbers of radiotherapy (RT) plans (between 250 and 600 radiotherapy plans) that did not allow the robustness of the models to be checked on a large set of plans. A study by Wang and colleagues [14] used an autoencoder deep learning-based model with 426 RT plans for training and 150 RT plans for validation. The results obtained show that a model performs better with 100% sensitivity and 83% specificity for technical validation and 100% sensitivity and 72% specificity for clinical validation.

However, these results are not satisfactory enough for routine clinical implementation, as the number of errors is still large, especially for complex RT plans (related to high number of arcs, high variation of flow, speed of the arm of the accelerator, speed of the leaves, to the MLC field opening…). Moreover, no public solution was developed from these models, which do not enable use in clinical practice. For these reasons, more complex AI models are needed for complex RT plans, and easy to use solutions available for everyone are needed to be clinically usable.

Subsequently, more complex and innovative AI models have been developed including the combination of machine learning and deep learning approaches, named deep hybrid learning (DHL). One example application of this method was to automate the analysis of radiographic images in order to diagnose various pathologies [15]. Today, to the best of our knowledge, no study has used the DHL model for the prediction of QA outcome on many RT plans.

The objective of this study was to develop AI models that are more complex than those proposed in the literature but still being clinically applicable. The models are based on the complexity indices of the treatment plans in order to predict the results of QA. The first step of this work was to create machine learning classification models for easy to moderately complex RT VMAT plans. In the second step, more complex and innovative DHL models have been developed for more complex RT VMAT plans. Finally, to use these models in clinical routine, a freely online platform has been developed.

## 2. Materials and Methods

### 2.1. Patient Cohorts

This retrospective study was approved by the local institutional review board. One thousand, six hundred and thirty-two patients with cancer referred to our oncological center between June 2019 and March 2022 were included. The Declaration of Helsinki and MR-004, a national French institution (INDS) defining health research conduct guidelines, were used for this study. The number of tumors per location is represented in Table 1. As several locations have low numbers of cases (digestive, rachis, other, member and skin tumor location) which is too low to build efficient models for these locations, we pool these cases in the “all” model only, and we did not develop specific models for these locations.

### 2.2. Radiotherapy Plan

Radiotherapy treatment using volumetric modulated arc therapy (VMAT) was undertaken with a RapidArc (Varian^®^, Le Plessis-Robinson, France) machine. Radiotherapy plans (RT-plan) were computed using Eclipse^®^ TPS. Complexity indexes were obtained from RT-plan using Raystation^®^ TPS and an in-house python code available at: https://github.com/AurelienCD/DeepHybridLearning_RadiotherapyQA_Depository_ManuscriptID_Diagnostics [16] and “Complexity_index.py” (accessed on 1 February 2023). To achieve an easy solution applicable to routine clinical settings, only six complexity indexes were used: Leaf Sequence Variability (LSV), Aperture Area Variability (AAV), Modulation Complexity Score (MCS), Leaf Travel (LT), combination of LT and MCS (LTMCS) and Small Aperture Score (SAS10) [5]. Complexity indexes were extracted from the plan and not from each arc.

### 2.3. Patient Specific Quality Assurance

The Delta4^®^ phantom from Scandidos^®^ was used for patient-specific quality assurance. To establish the QC conformance, the local gamma index has to be higher than 0.95% with 3%/3 mm and the gamma mean needs to be below a threshold (Figure 1).

Using significantly (*p* < 0.05) different gamma mean values, a gamma mean threshold was established for each tumor location using the following formula:GM_threshold_ = x + 1.96 × σ
where:
x: gamma mean valueσ: standard deviation

The GM_threshold_ value for each tumor location is summarized in Appendix A. To compute AI predicting models, conformance and non-conformance QC was assigned to 1 and 0, respectively.

### 2.4. Artificial Intelligence Algorithms

#### 2.4.1. Machine Learning (ML)

Input data of the ML models were six quantitative variables obtained from the RT plans which represents the complexity of the radiotherapy treatment. LSV, AAV, MCS, LT, LTMCS and SAS10 are sensitive to the variation of the leaf position, number of monitor unit and area of irradiation. The aim of the ML models was to be able to predict, from these six complexity indexes as input, the two classes of conformance and non-conformance QC (0 or 1, respectively) as output. ML classification models were used for the low to medium complexity RT plans of tumors localized in the brain and thorax. The models were developed with hyperparameter optimization and cross-validation using the Sklearn python library [17]. This library provides a selection of efficient tools for ML and statistical modeling including classification, regression, clustering and dimensionality reduction. Training and testing were proportionally assigned to 80% and 20%, respectively. Several ML models were tested and compared by evaluating their prediction scores: Linear Discriminant, Linear Regression, Ridge, Gaussian NB, Decision Tree, Support Vector Classifier, Stochastic Gradient Descent and Random Forest Classifier. ML model hyperparameters were optimized using gridsearch and the cross-validation tools of the Sklearn library. As presented in the code available online, for each ML model classifier, the hyperparameters were tuned to obtain a higher validation score and area under the curve (AUC).

Following model optimization, Receiver Operating Characteristic (ROC) curves and confusion matrices were computed for the best models. This was conducted to find the optimal probability threshold for obtaining as few as possible false positives. All the ML codes with detailed documentation are available at: https://github.com/AurelienCD/DeepHybridLearning_RadiotherapyQA_Depository_ManuscriptID_Diagnostics and “Machine Learning.ipynb” (accessed on 1 February 2023).

#### 2.4.2. Deep Hybrid Learning (DHL)

RT treatment plans for pelvis, breast and H&N tumor locations are generally more complex than those of brain and thorax tumor locations. For example, VMAT plans treating breast tumor location have to manage build-up issue, the difference between tissue (proximity of lung and breast tissues), beam entry surface is not planar, etc. For this reason, more complex AI models were needed and deep hybrid learning involving the combination of ML and DL was used. As shown in Figure 2, the ML models described in the previous section have as input six quantitative values of complexity indexes, and each ML model predicted a probability (between 0 to 100%) of conformance or non-conformance QC. The probabilities of non-conformance QC of the ML models which have more than 80% prediction score were then given to a multilayer deep learning (artificial neural network) model to obtain probabilities of conformance and non-conformance QC. The deep learning model was computed using the Tensorflow (v2.8.0) and its Keras library [18]. Optimization of the hyperparameters such as the activation function, number of layers, number of neurons in each layer, loss function, regularizer function, learning rate, metrics and percentage of dropout was performed using hyperparameter tuning with KerasTuner.

This optimization gave 2 dense layers of 658 neurons and an output layer of two neurons with an L2 regularization function and a learning rate of 0.005.

The activation function and dropout were relu, 0%; selu, 60% and softmax in order. The model was built with a binary cross-entropy loss function and binary accuracy optimization metrics on 100 epochs. As with the majority of medical datasets, our data were imbalanced with an important proportion of conformance QC in comparison to non-conformance QC; to avoid the negative impact of this characteristic, the Imblearn python library and Keras class weight was used [19]. In addition, to manage the imbalance data, weights of the model were adjusted regarding the proportion of classes.

All the DHL codes with detailed documentation are available at: https://github.com/AurelienCD/DeepHybridLearning_RadiotherapyQA_Depository_ManuscriptID_Diagnostics and “Deep Hybrid Learning.ipynb” (accessed on 1 February 2023).

### 2.5. Statistical Analysis

The performance of the ML and DHL models was evaluated considering the accuracy score of Sklearn and Tensorflow of the test dataset, area under the curve (AUC) of the ROC curves, specificity, sensitivity and finally the number of false positives. All statistical analyses were completed in python [20] using the statistical data visualization and Seaborn library [21,22].

## 3. Results

### 3.1. Prediction Models for All Tumor Location

The first approach was to develop general machine learning models able to predict QC conformance whatever the tumor location. The following ML models were used. As shown in Figure 3, the large number of RT plans used for ML resulted in good training and validation scores for the different ML models used. Random Forest Classifier training and validation scores were the highest with 100% and 89.9% respectively. Quantitative performances of the ML models classifier are presented in Appendix A.

A confusion matrix is used to visualize the performance of the Random Forests ML algorithm with a summary of the errors given in Table 2. It is noteworthy that only 1.10% false positive (FP) values were obtained. ROC analysis of only 18 RT plans using the Random Forests ML algorithm gave an AUC value (0.97) close to 1 (Figure 4 and Table 2).

Despite these good results, application of the general ML models to specific tumor locations did not provide acceptable results for clinical practice. For example, application of the Random Forests Classifier model for brain tumor RT plans gave a 4.49% FP rate and specificity of 30%; see Appendix A and Appendix A. Therefore, ML models for each tumor location had to be developed to increase the performance of the models.

### 3.2. Prediction for Brain and Thorax Tumor Location: Machine Learning Models

The specific ML for each tumor location gave better results than the model developed on all the tumor locations. For brain tumor RT plans, perfect prediction was obtained with zero FP, resulting in an AUC equal to one and an accuracy and specificity of 100%; see Appendix A and Appendix A. Similar results were obtained using the specific ML model for thorax tumor RT plans. However, for more complex RT plans such as the pelvis, breast and H&N tumors, ML models were not accurate enough. As an example, ML models for breast tumor RT plans gave a specificity of only 87%. For this reason, more complex AI models using deep hybrid learning (DHL) were developed for pelvis, breast and H&N tumors.

### 3.3. Prediction for Pelvis, Breast and H&N Tumor Location: Deep Hybrid Learning Models

DHL models presented very accurate results. As an example, the DHL model for breast cancer reduced the number of FP from 9 with ML to 0, increasing the specificity to 97.7% (87% with ML only model); see Figure 5 and Table 3.

Similar results were found with DHL models for H&N tumor locations (FN of ML and DHL model for H&N location = 5.88% and 1.47%, respectively).

As previously shown, AI models had to be adapted regarding the tumor location which involve more or less complex RT dosimetry plans. The performance and architecture of each model for the different tumor locations are summarized in Table 4 below:

### 3.4. Application of the Solution in Clinical Practice

The prerequisite of this study was to develop a facile approach that could easily be implemented in clinical practice. For this reason, an application programming interface (API) was developed which can easily be used in a clinical environment. Six complexity indexes must be entered in the user interface and the tumor location chosen; then, ML or DHL models are used to predict patient-specific QC conformance. The API is freely available at: https://aureliencd-radiotherapy-quality-api-cq-patient-predictor-2clxve.streamlitapp.com/ (accessed on 1 February 2023). The performance of the API was only optimized on our data and needs to be fully validated before external use. The mean time (minute) taken for calculation of the complexity index, and conformance prediction with the platform, was five min per plan. If we only measure the plans with a non-conformance prediction, the time saved per week is around 135 min compared to the 150 min normally taken to perform patient-specific QA, which gives the possibility of treating seven more patients. The complete workflow we proposed in our center to use this solution is presented in Appendix A.

## 4. Discussion

Patient-specific assurance control in RT is mandatory but very time-consuming and consequently impacts RT machines’ availability for patient treatment [23]. As QC results are highly correlated with the complexity of the RT dosimetry plans, we proposed the use of prediction models using RT plans complexity indexes as input to predict QC outcome while guaranteeing treatment assurance [2]. In this study, we firstly used a similar approach to that described in the literature by thresholding complexity indexes to predict QC outcome. As shown in Appendix A, our results are similar to those of Jazouli and co-workers [9], as we obtained a high number of TP; however, the FN rate was significant and correlated with the complexity of the RT plan. We secondly developed regression models to predict absolute gamma index values. Valdes and colleagues [24] showed good results using this approach with predicting errors smaller than 3%; however, in our study, ML regression models resulted in validation scores around 0.3; see Appendix A. For this reason, we expanded to classification ML models. A recent study by Granville and colleagues, in a large cohort similar to that of our study, reported an AUC of 0.80 to 0.92 for mixed tumor locations using a support vector classifier [12]. Our classification ML model for easy RT dosimetry plans (brain and thorax tumors) yielded better results with AUC = 1 and 0.99 for brain and thorax RT plans, respectively. As noted in a recent review by Simon and colleagues [25], Granville and now our study are the only studies that have developed ML models using a large number of patients with VMAT RT plans. Due to the more complex technique used for VMAT in comparison with intensity-modulated radiotherapy (IMRT), developing a prediction solution for QA is highly relevant. To the best of our knowledge, our study is the first using the innovative deep hybrid learning approach for patient-specific QA prediction. For a moderately complex dosimetry plan (prostate/pelvis tumors location), a more complex AI DHL model was required. In the literature, Kimura and colleagues developed a deep learning solution for the QA of prostate plans and obtained a specificity value of 0.986 [26], which is in agreement with that obtained in our study (specificity value for pelvis tumor location = 1). However, no study has been published for complex VMAT dosimetry plans (breast or H&N). Interian and colleagues used a deep learning approach for patient-specific QA prediction mainly based on breast tumor plans and obtained a mean absolute error of 0.70 [27]. We obtained similar results with our ML-based model for breast tumors plans with a specificity of only 87%. We therefore developed a more innovative model using DHL to increase the specificity to 97.7%.

Most of the literature uses gamma index metrics to determine patient-specific QA conformance. In this study, we chose to use a combination of gamma mean and gamma index to provide a more patient-specific QC result more similar to that of our clinical practice. Whereas both parameters give complementary information, to the best of our knowledge, this is the first study developing an AI model using both parameters. As it is mentioned by the Task group 218 of the American Association of Physicists in Medicine [28], a gamma rate of 95% with 3%/3 mm is the most used criteria in the world even if they proposed to be more specific regarding the type of treatment and for example use 3%/2 mm. As the gamma rate of 95% with 3%/3 mm is still the most used in the world, we decided to develop the AI model using this criterion.

When determining the features for AI modeling, it is important to select which complexity indexes are relevant and should be incorporated in the model. Granville and colleagues highlighted the importance of including the number monitor unit (MU) in their model. As it can be observed in our code online, the six complexity indexes used in our study accounted for the MU; however, when we analyzed the importance of specific features, no differences were observed between the six complexity indexes.

Several authors have used the log file from machine and MLC to predict QA conformance [29,30,31]. These approaches gave interesting results, and it would be interesting to combine complexity indexes and log file approaches to improve the efficacy of the AI prediction model. However, these approaches need to use the machine to obtain the log files or have to be completed after patient treatment as post-treatment patient-specific QA.

As QA of the prediction model itself, we add to the annual QC of the TPS a QC of the script used in routine for complexity indexes extraction and prediction constancy for the API. Ten constancy plans for different tumor locations are used for this control: the script is run, the given metrics are compared with the reference (i.e., metrics given at the first run of the script) and the prediction conformance of the QC is also compared to the reference prediction.

A significant number of studies have developed AI models to accelerate the patient QA process in RT. However, very few studies have implemented the models clinically. A study was implemented clinically, but the results obtained for the AUC were at best 0.869 [32]. Our initial aim of this study was to develop an AI solution for QC in VMAT that is easily usable by everyone and can be implemented in clinical routine. For this reason, we used only six complexity indexes in the modeling, which can readily be obtained during the dosimetry step while packaging the solution in an API for ease of use by all.

The reproducibility of AI models in different clinical settings and in different centers is key to its clinical implementation. By using a federated learning approach, multi-centric AI models led to the development of a global model based on local model updates [33]. Our model was developed at a single medical center; the next step is to use a federated learning approach with voluntary centers to improve our model and increase its relevance for other centers.

## 5. Conclusions

As patient-specific QA takes time on the treatment machine with the impossibility to treat patients during that time, the objective of this study was to propose an alternative solution for patient-specific QA that would make treatment machines more available to patients. We developed AI models based on complexity indexes to predict patient-specific QA conformance for VMAT treatments. We show that the ML models used were able to predict with 100% specificity the QA results for easy dosimetry plans for brain and thorax tumor location. For moderate to more complex dosimetry plans of tumors located in the pelvis, H&N and breast, innovative DHL models were required, providing specificity of 84% to 100%. In our clinical practice, this solution leads to use the solution in that way: before patient-specific QA, if the result is okay, the QA would not be completed. However, if the result is not okay, patient-specific QA is performed, and if it is still not okay, then the RT plan is re-computed with lower complexity. All these processes are under the responsibility of the medical physicist.

We further developed an API that is freely available to the scientific community for the application of the QA solution in clinical routine. Subsequently, we will be able to propose transfer learning or federate learning solutions so that our solution can be used and optimized in other partner radiotherapy centers

## Figures and Tables

**Figure 1 diagnostics-13-00943-f001:**
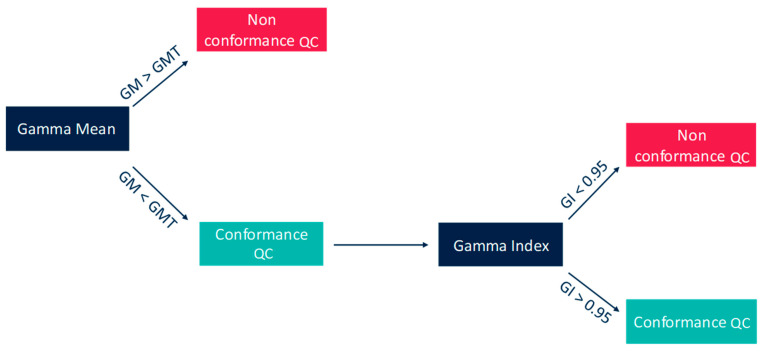
Decision tree of the QC conformance using gamma index and gamma mean.

**Figure 2 diagnostics-13-00943-f002:**
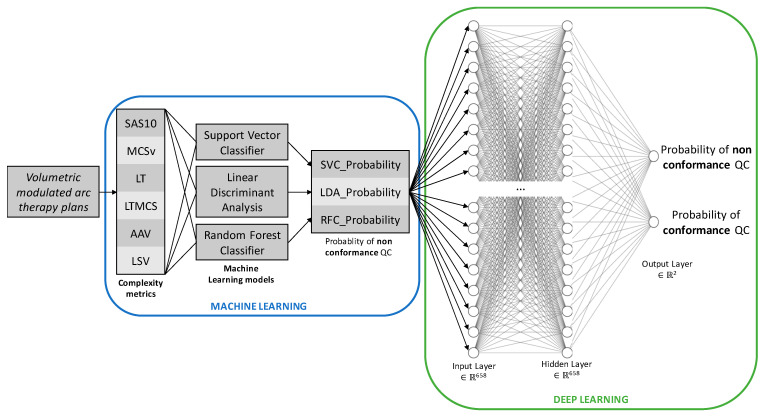
Deep hybrid learning model architecture (breast tumor location example).

**Figure 3 diagnostics-13-00943-f003:**
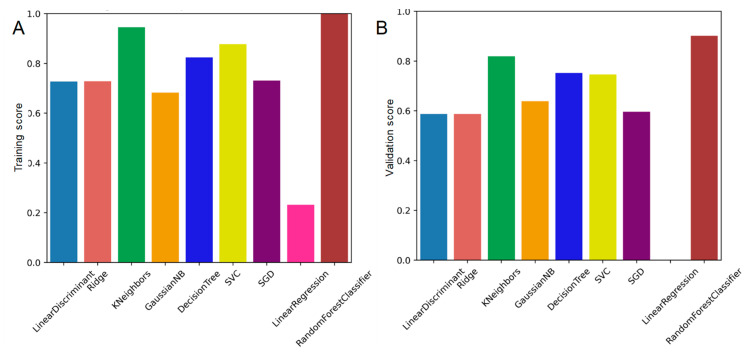
ML model performance for all tumor locations. Training score (**A**) and validation score (**B**).

**Figure 4 diagnostics-13-00943-f004:**
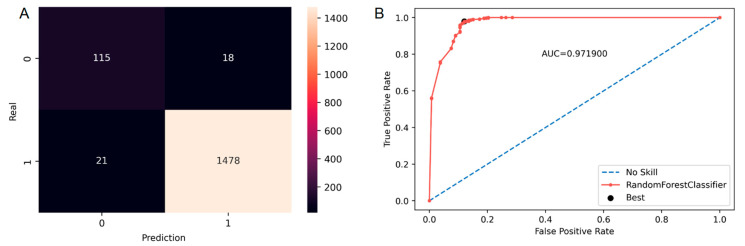
Confusion matrix (**A**) and ROC curve (**B**) for all tumor location ML models using Random Forest Classifier.

**Figure 5 diagnostics-13-00943-f005:**
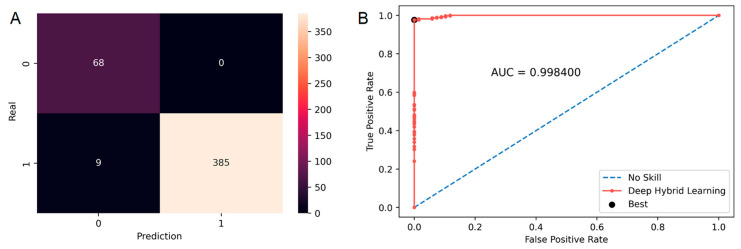
Confusion matrix (**A**) and ROC curve (**B**) for breast tumor DHL model.

**Table 1 diagnostics-13-00943-t001:** Description of the patient cohort and tumor location.

Tumor Location	Number of VMAT Plans
Pelvis	576
Breast	462
H&N	204
Brain	156
Thorax	99
Digestive	49
Rachis	30
Other	25
Member	17
Skin	14
**All**	**1632**

**Table 2 diagnostics-13-00943-t002:** Performance for all tumor location ML models. TP = true positive, TN = true negative, FP = false positive and FN = false negative.

TP	TN	FP	FN	Sensitivity	Specificity
90.56%	7.05%	1.10%	1.29%	98.6%	86.47%

**Table 3 diagnostics-13-00943-t003:** Performance for breast tumor DHL model.

TP	TN	FP	FN	Sensitivity	Specificity
83.33%	14.72%	0%	1.95%	97.7%	100%

**Table 4 diagnostics-13-00943-t004:** AI models architecture and performance for each tumor location.

Locations	AUC	Sensitivity	Specificity	Accuracy	TP	TN	FP	FN	Architecture
Brain	1	100%	100%	100%	94.87%	5.13%	0%	0%	ML
Thorax	0.9986	98.90%	100%	98.99%	90.91%	8.08%	0%	1.01%	ML
Pelvis	0.9869	100%	90%	99.65%	96.53%	3.13%	0.35%	0%	DHL
Breast	0.9984	97.72%	100%	98.05%	83.33%	14.72%	0%	1.95%	DHL
H&N	0.9589	98.32%	84%	96.57%	86.27%	10.29%	1.96%	1.47%	DHL
All	0.9891	99.33%	98.50%	97.43%	91.24%	8.03%	0.12%	0.61%	DHL

## Data Availability

The data presented in this study can be sent upon reasonable request and the python code used are openly available at https://github.com/AurelienCD/Radiotherapy_Quality_Control_API (accessed on 1 March 2023).

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
