# Peer review of "Deep Hybrid Learning Prediction of Patient-Specific Quality Assurance in Radiotherapy: Implementation in Clinical Routine"

_diagnostics, 2023, doi:10.3390/diagnostics13050943_

Round 1
Reviewer 1 Report
Your paper is very interesting, but there is some issues you have to be concern:
1. describe the nature of the data.
2. The available data is imblanace for all classes so you have to choose the lowest number and randomly select the same number of records from the whole data to balance and make your model meaningful.
3. I think you can drop some low number of cases recorded and preserve the rest to build a solid model.
4. describe the model of deep learning and show its details
5. I think you use majority voting is that correct
6. You did not explain anything about machine learning classifiers that have been used beside their results
7. Overall your paper needs more clarification
Author Response
We thank the reviewer for his comments.
The manuscript was revised in accordance to the reviewer’s comments as far as possible and answers to comments raised have been addressed in a point-by-point manner in the file attached

Reviewer 2 Report
The manuscript titled “Deep hybrid learning prediction of patient-specific quality assurance in radiotherapy: implementation in clinical routine” seems interesting. But there are some changes that need to be incorporated.
1. The literature has some old references, more recent literature is recommended.
2. The paper refers to the term “accuracy” in the abstract where no accuracy is computed.
3. Add evaluation parameters such as accuracy.
4. Comparison with state-of-the-art techniques is recommended.
5. Conclusion section needs to be enhanced.
6. Add a table in the results section to show the performance of all classifiers for a better understanding of the reader.
Author Response
The manuscript was revised in accordance to the reviewer’s comments as far as possible and answers to comments raised have been addressed in a point-by-point manner in the file attached.

Round 2
Reviewer 1 Report
It can be accepted
Author Response
Dear Reviewer,
First of all I apologyze for the late answer, I was on holiday without access to my emails.
Please find two sentences we have added in the discussion as asked by the academic editor :
1)
"In our clinical practice this solution leads to use the solution in that way: before patient specific QA, if the result is ok the QA would not be done, however if the result is not ok, patient specific QA is performed and if it is still not ok, RT plan is re-computed with lower complexity. All these process are under the responsibility of the medical physicist."
2)
"As patient specific QA takes time on the treatment machine with the impossibility to treat patients during that time, the objective of this study was to propose an alternative solution for patient specific QA that would make treatment machines more available to patients."
